# SPARQL hash functions as pipeline triggers

Damien Graux[1,2]([✉]) [ID], Fabrizio Orlandi[2]([✉]) [ID], and Declan O'Sullivan[2] [ID]

[1] Inria, Université Côte d'Azur, CNRS, I3S, France
[2] ADAPT SFI Centre, Trinity College Dublin, Ireland
{grauxd,orlandif,declan.osullivan}@tcd.ie

**Abstract.** The recent increase of RDF usage has witnessed a rising need of "verification" around data obtained from SPARQL endpoints. It is now possible to deploy Semantic Web pipelines and to adapt them to a wide range of needs and use-cases. Practically, these complex ETL pipelines relying on SPARQL endpoints to extract relevant information often have to be relaunched from scratch every once in a while in order to refresh their data. Such a habit adds load on the network and is heavy resource-wise, while sometimes unnecessary if data remains untouched. In this poster, we present a useful method to help data consumers (and pipeline designers) identify when data has been updated in a way that impacts the pipeline's result set. This method is based on standard SPARQL 1.1 features and relies on digitally signing parts of query result sets to inform data consumers about their eventual change.

## 1 Introduction

During the past decades, the number of linked open datasets has rapidly increased[1]. These datasets are structured following the W3C standard Resource Description Framework (RDF) [6] and share knowledge on various domains, from the generalist ones such as DBpedia [1] or WikiData [7] to the most specialised ones, *e.g.* SemanGit [5]. This abundance of open datasets and SPARQL [2] endpoints led not only researchers but also businesses to integrate RDF graphs into their complex data pipelines. In particular, businesses are increasingly leveraging Semantic Web technologies to structure their own data and create value, sometimes integrating external Linked Data to enrich their analyses [4].

Benefiting from the two decades of developments made by the community, it is now possible to deploy Semantic Web pipelines and to adapt them to a wide range of needs and use-cases. Recent developments have been, for example, focused on distributed systems or on connecting Semantic Web data management systems together with non-RDF centric systems, paving the road to querying heterogeneous data. As a consequence of this increasing complexity of the use-cases, the pipelines themselves are getting more complicated, and often rely on several distinct data sources in order to compute their final results.

---

[1] From 2010 to 2020, the LOD-cloud has grown from 203 to 1 255 datasets, approximately: https://lod-cloud.net/

Hence, as data available may change, these pipelines (or parts of them) are frequently re-run in order to get fresher results. However, lots of times they are re-run unnecessarily as datasets have not been updated in the meantime in ways that impact the result sets of the pipeline. All these operations are leading to a waste of computation power and loads on the network.

In this poster, mainly dedicated to SPARQL practitioners and data pipeline designers, we review the possibilities provided by the SPARQL 1.1 standard [2] to sign query result sets. In particular, we will discuss how these methods can be used to optimise data pipelines avoiding expensive re-computation of results when data triples have not been updated.

## 2  SPARQL 1.1 hashing capabilities

The SPARQL standard provides a large set of built-in functions, from ones dedicated to strings to specific ones about dates. These can be used by query designers to refine their result set. In particular, the standard offers a set of five hash functions[2]: MD5, SHA1, SHA256, SHA384 & SHA512.

General signature of the hash functions:

```
simple literal  hash_function (simple literal arg)
simple literal  hash_function (xsd:string arg)
```

Example using MD5:

```
H = md5("ab") =  md5("ab"^^xsd:string)
H = "187ef4436122d1cc2f40dc2b92f0eba0"
```

These functions accept either RDF literals or strings as argument and return the hash as a literal. In addition, a `xsd:string` or its corresponding literal should return the same result. In the 'MD5' example above, the hash value represents the result of a simple SPARQL query[3].

Practically, these functions can be used to hash a complete RDF graph accessible through a SPARQL endpoint. Indeed, one can extract all the triples available with `select * where {?s ?p ?o}`, and then hash all of them, aggregated with a `group_concat` function. This could look like so:

```
SELECT (SHA1(GROUP_CONCAT(?tripleStr ; separator=' \n'))) AS ?nTriples
WHERE { ?s  ?p  ?o
   BIND(CONCAT(STR(?s), " ", STR(?p), " ", STR(?o)) AS ?tripleStr) }
```

In the previous query, the triples `?s ?p ?o` are cast by element to a string (`STR`), and then concatenated to form a "triple". The recomposed list of triples is then grouped into one single string (`GROUP_CCONCAT`) and finally hashed.

Although easy to understand, this "naïve" approach has some drawbacks. *First*, the result depends on the order of the triples returned by the triplestore: a workaround can be achieved adding *e.g.* `ORDER BY ?s ?p ?o datatype(?o) lcase(lang(?o))`. *Second*, this method has a scalability issue, as all the graph is loaded in-memory before the hash call. We therefore recommend this approach to sign small RDF graphs, *e.g.* ontologies or small result sets. *Finally*, this method

---

[2] https://www.w3.org/TR/sparql11-query/#func-hash

[3] `select * where{ values ?x {"ab" "ab"^^xsd:string} bind (md5(?x) as ?H)}`

does not address the complex case of blank node identification as *e.g.* {_:a p o} and {_:b p o} do not have the same hashes (see *e.g.* [3] for algorithmic solutions).

## 3    Tracking result updates of SPARQL queries

Signing an RDF graph through a SPARQL query is not as reliable as the traditional and complete method that transforms the entire graph beforehand. However, it allows users to compare different query results for the same query on the same engine. As we know, on the same endpoint, the same query (without calls to functions like RAND or NOW) is supposed to return the same result set for the same dataset. Therefore, we think this SPARQL-based "lightweight" signing approach could be useful for ETL pipeline designers.

Indeed, a common challenge for pipeline designers is to know when a refresh (*i.e.* a re-run, often from scratch) is needed, following a data update. Often, there is no way to know *a priori* that datasets have been updated and, thereby, pipelines are often run even when nothing has been modified. This, unfortunately, leads to time-consuming and (sometimes) costly processes in terms of both resources and network bandwidth, as multiple intermediate results involved by the pipelines are shuffled.

We suggest to use the aforedescribed approach to check on the endpoint side if the results of a SPARQL query have changed. A hash of the results could be computed by the endpoint and be compared with a previously obtained one. In case of a mismatch, the query (and the rest of the pipeline) could be run again. Assuming Q is the considered SPARQL select query, we propose the following steps to generate the query which computes the hash of the results of Q:

1. Extract and sort the list of distinguished variables V (if a * is given, the considered variables are the ones involved in the where);
2. Wrap Q in a select * query ordered by V;
3. Embed the obtained query in a select query computing the hash of the grouped concatenation of the cast (to string) distinguished variables.

To give an example, if we consider the query which extracts from DBpedia the current members of English-named Punk_rock groups, Q=

```
SELECT ?members ?bandName WHERE {
 ?band dbo:genre dbr:Punk_rock . ?band dbp:currentMembers ?members .
 ?band foaf:name ?bandName FILTER(langMatches(lang(?bandName), "en")) }
```

Its sorted list of distinguished variables would be ?bandName ?members. And to obtain a (MD5-)hash of the results of Q, we should run:

```
SELECT MD5(GROUP_CONCAT(CONCAT(STR(?bandName),STR(?members)); separator=' \n'))
as ?H WHERE {
 SELECT * WHERE {                          # Collecting all the ordered results
  SELECT ?members ?bandName WHERE {     # The original query
   ?band dbo:genre dbr:Punk_rock . ?band dbp:currentMembers ?members.
```

```
   ?band foaf:name ?bandName FILTER(langMatches(lang(?bandName), "en")) }
 } ORDER BY ?bandName ?members          # Ordering by distinguished variables
}
```

The three steps to generate the query[4] that obtains the hash are easy to automate and allow users to know when to relaunch their pipelines. All this while making as much computations as possible on the endpoint side in charge of computing the hash.

## 4   Conclusions

This poster paper describes how to improve existing Semantic Web data pipelines with a SPARQL-based method that helps in identifying when query results have changed. It allows to re-run pipelines only when interesting parts of the original datasets have been updated. By using SPARQL to compute the signature of the query results, it avoids large result sets to be sent over the network. We hope this will inspire developers to use the hash functions provided by the standard, and serve our method at: https://dgraux.github.io/SPARQL-hash/ where our query converter can be used directly by developers to generate queries computing the hash of their result sets.

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
