# OpenReview forum: "SPARQL hash functions as pipeline triggers"
_eswc-conferences.org/ESWC/2021/Conference/Poster_and_Demo_Track — Submitted to ESWC2021 P&D_

### Official Review · AnonReviewer4 · 2021-04-12
**Limited contribution, literature review and future directions missing.**

**Rating:** 3
**Confidence:** 4

**Review:**

The authors propose to use SPARQL hashing functions over the concatenation of string representation of query results in order to know if a dataset has changed, arguing this would improve ETL pipelines.

I find that the proposal is not groundbreaking, and the benefits offered by the solution are not clear, since there is no evidence or experiments provided. Furthermore, there isn't a review of the relevant literature needed to grasp what the current solutions are, if any, and how this proposal would improve upon that or how it would compare to other approaches.

The paper indicates three drawbacks of the proposal, giving a solution only to one of them, and not discussing enough how the second (scalability) and third (blank nodes) could be solved.

**Anonymity:**

Yes, I would like my review to remain anonymous.

---

### Official Review · AnonReviewer1 · 2021-04-13
**Inspiring strategy, probably missing some proof**

**Rating:** 6
**Confidence:** 4

**Review:**

The paper proposes a strategy for detecting interesting modifications in a graph, using the SPARQL 1.1 hash function.

The work is well motived, also proposing in Section 2 some not optimal solutions.
However, it is not clear how to manage:
- the choice of the query _Q_. A wrong choice may lead to not finding important modifications and not re-run the pipeline.
- the scalability issues underlined in Section 2. Potentially, a large part of the graph can be selected and kept in memory also in this case.

The paper does not present applications in a real environment, nor propose any sort of evaluation.

The strategy is anyway inspiring and both a repository and a demo page is available.

**Anonymity:**

Yes, I would like my review to remain anonymous.

---

### Official Review · ~Ben_De_Meester1 · 2021-04-13
**Too trivial description of a SPARQL hash function**

**Rating:** 3
**Confidence:** 4

**Review:**

This poster paper describes a, in my opinion, quite trivial way to hash a SPARQL result set. Discussion on scalibility and alternatives is limited, which makes me doubt the usefullness of publishing this as an academic paper, certainly since it does not seem to add much more than a tweet from Martynas: <https://twitter.com/namedgraph/status/1351521880369491968>

- "In this poster, mainly dedicated to SPARQL practitioners and data pipeline designers" -> I really like the fact that you set the stage
- Are language and datatype taken into account or not? Are duplicates taken into account or not? ie what if I have an rdf graph diff where one literal has either no datatype specified or xsd:string specified? What effect does this have on the pipeline?
- What is the advantage compared to just doing a hash from the results, outside of SPARQL? Doesn't this give more flexibility and options to optimize?
  - What about pipelines that can cache intermediate results? Isn't thatm more generic that this thing that only works on SPARQL endpoints?
  - How costly is the order by clauses? It's all about small SPARQL query results, so how do you know this is better than the alternative of just re-running the same pipeline?
- "By using SPARQL to compute the signature of the query results, it avoids large result sets to be sent over the network" (conclusion) directly contradicts with "We therefore recommend this approach to sign small RDF graphs, e.g.ontologies or small result sets" (section 2)
- Your online demo makes it seem as if any hash function is OK: are there any upsides or downsides about these hash functions? I have the feeling it's easiest to just use the least costly hash function, why is that feeling wrong?
- I would suggest providing the JavaScript implementation as a library instead of (only) the online demo: I think that would help developers much more to integrate your method than a GUI.
- Of course it's impossible to tell, but what about <https://twitter.com/namedgraph/status/1351521880369491968> ? How can you argument that this work is yours and worthy of an academic publication, since it doesn't seem to contain more investigation than this tweet?

**Anonymity:**

No, I would like my review to be deanonymized.

---

### Decision · Program_Chairs · 2021-04-19

Reject